# Soy Milk Consumption in the United States of America: An NHANES Data Report

**DOI:** 10.3390/nu15112532

**Published:** 2023-05-29

**Authors:** Maximilian Andreas Storz, Maria Brommer, Mauro Lombardo, Gianluca Rizzo

**Affiliations:** 1Department of Internal Medicine II, Centre for Complementary Medicine, Medical Center—University of Freiburg, Faculty of Medicine, University of Freiburg, 79106 Freiburg, Germany; 2Interdisciplinary Medical Intensive Care (IMIT), Medical Center—University of Freiburg, Faculty of Medicine, University of Freiburg, 79106 Freiburg, Germany; 3Department of Human Sciences and Promotion of the Quality of Life, San Raffaele Roma Open University, 00166 Rome, Italy; mauro.lombardo@uniroma5.it; 4Independent Researcher, Via Venezuela 66, 98121 Messina, Italy; drgianlucarizzo@gmail.com

**Keywords:** soybean milk, plant-based milks, milk substitutes, soy, consumption, consumer attitudes, prevalence, NHANES

## Abstract

With the increasing adoption of plant-based diets in the United States, more and more individuals replace cow milk with plant-based milk alternatives. Soy milk is a commonly used cow milk substitute, which is characterized by a higher content of polyunsaturated fatty acids and fibers. Despite these favorable characteristics, little is known about the current prevalence of soy milk consumption the United States. We used data from the National Health and Nutrition Examination Surveys (NHANES) to assess soy milk usage in the United States and identified potential predictors for its consumption in the US general population. The proportion of individuals reporting soy milk consumption in the NHANES 2015–2016 cycle was 2%, and 1.54% in the NHANES 2017–2020 cycle. Non-Hispanic Asian and Black ethnicities (as well as other Hispanic and Mexican American ethnicities in the 2017–2020 cycle) significantly increased the odds for soy milk consumption. While a college degree and weekly moderate physical activity were associated with significantly higher odds for consuming soy milk (OR: 2.21 and 2.36, respectively), sex was not an important predictor. In light of the putative health benefits of soy milk and its more favorable environmental impact as compared to cow milk, future investigations should attempt to identify strategies that may help promote its consumption in selected populations.

## 1. Introduction

The plant-based diet is increasingly adopted by the general population in Western countries and has also attracted the interest of the scientific community and the food industry [1,2,3]. As a result, the market has increased the available amounts of innovative plant-based foods to meet this growing demand [4,5]. The interest in switching to plant-based alternatives is frequently derived from ethical aspects and advantages associated with health [6,7], and recently also from a greater sensitivity towards environmental aspects that have emerged from the scientific literature [8,9,10].

Adoption of vegetarian and vegan diets has shown a beneficial effect on cancer incidence [6], and has been associated with a reduction in cardiovascular morbidity and mortality in recent clinical studies [7,11]. These aspects are particularly relevant considering that around one third of cardiovascular and neoplastic diseases in the world could be prevented by increasing fruit and vegetable intake, according to the World Health Organization and the World Cancer Research Fund [12].

With the increase in the demand for plant-based foods, the consumption of alternatives to cow milk also raised, with a forecast increase of over 10% from 2000 to 2024 (globally), with the major trend observed for the Asia-Pacific region [13]. At the same time, research has also moved to bridge the gap between consumer needs (milk allergy, lactose intolerance, or vegan diet) and commercial options [14,15,16]. Although the term “milk” had already been regulated as an exclusive term for the mammary secretion of cows and other mammals by the Food and Drugs Administration (FDA) and the European Union [17,18], the FDA recently issued a recommendation regarding the labelling of plant-based dairy alternatives, defining the lawfulness of including the term “milk” [19]. The consumer is now thoroughly familiar with these foods so there is no longer need for the previous terminological restrictions, with the recommendation of clear labelling regarding the nutritional properties of the products. Accurate labelling and fortification of plant-products already available on the market would allow consumers to assess the adequacy of vitamins and other micronutrients usually lacking in these products if compared to cow milk [20].

Among plant-based drinks, one of the most commonly used as a substitute for cow milk is soy milk [21]. Soy is a widely used food in vegetarian diets [22]. Among its nutritional characteristics, soy milk is the only plant-based alternative to cow milk with a similar protein content [23]. Furthermore, it has a comparable Digestible Indispensable Amino Acid Score, demonstrating a good protein quality [24]. Additionally, soy milk is characterized by a higher content of polyunsaturated fatty acids, fibers, and by the absence of cholesterol [25]. These features may help reduce LDL levels [26]. The replacement of cow milk with soy milk could have an advantage in vegetarian diets as regards the absence of iron in the former and the possible presence of vegetable ferritin in the latter [27].

Soybean crops have a relevant environmental impact, with a variable effect on factors such as eutrophication, acidification, and global warming in different countries [28], and with a negative social impact on humans [29]. Nevertheless, soybean represents the main source of animal feed production [30]. Moreover, almost 80% of the world’s soy production is destined for livestock, including milk and dairy production [31], with about 2% designated for soy milk for humans [32].

Used as an alternative to cow milk, soy milk represents a more sustainable solution in terms of environmental impact and can be consistent with food security objectives [33]. Even if the presence of isoflavones has raised health concerns, it could have an advantage in mitigating menopausal disorders, without critical hormonal and fertility disturbance [34,35]. Nonetheless, soy milk has shown beneficial antioxidant actions, mainly attributable to the content of isoflavones [36].

Based on comments submitted to the FDA, dietitians appear to have a more accurate understanding of plant-based substitutes than other healthcare professionals [37]. More than half of consumers do not believe that dairy products are nutritionally better than plant-based alternatives and think that the latter can be part of a healthy diet [37]. In a sensory evaluation study, soy milk was shown to be the most popular milk alternative across various groups of participants, including omnivores and vegans [38].

Soy milk is one of the most common plant-based alternatives to cow milk and the only plant-based dairy substitute in the Dietary Guidelines for Americans [39]. Yet, data on its consumption in the US is sparse. This cross-sectional study sought to investigate the prevalence of soy milk consumption in a large and nationally representative cohort of American adults (NHANES—National Health and Nutrition Examination Survey) and aimed at a better understanding of its association with correlated sociodemographic aspects.

## 2. Materials and Methods

### 2.1. Study Population and Design

This analysis is based on data from the NHANES—an ongoing program of studies by the Centers for Disease Control and Prevention designed to comprehensively assess the health and nutritional status of the non-institutionalized U.S. population [40,41]. The NHANES’ complex multistage, stratified, clustered, and probability sampling design allows for nationally representative health and nutritional status assessments. Key program characteristics (including recruitment methods, study size, and study execution details) have been described elsewhere in detail [39,40]. NHANES was approved by the National Center for Health Statistics (NCHS) and all study participants gave written and oral consent to the study [42].

For this analysis, we used data from two different NHANES cycles: (I) the NHANES 2015–2016 cycle, and (II) NHANES 2017–2020 (which is also called the pre-pandemic cycle) [43,44]. Both cycles were analyzed independently for methodological issues because some important variables that were included in the 2015–2016 cycle were no longer available in the NHANES pre-pandemic cycle.

### 2.2. Primary Outcome Variable

Data on soy milk consumption was obtained from the NHANES Diet Behavior and Nutrition questionnaire. This module provides personal interview data on various dietary behavior and nutrition related topics. Amongst others, it includes one question on milk product consumption in the past 30 days. Said question reads as follows:


*“In the past 30 days, how often did you have milk to drink or on your cereal?”*


Participants were instructed to include chocolate and other flavored milks as well as hot cocoa made with milk. Moreover, they were instructed not to count small amounts of milk added to coffee or tea. The question did not cover milk usage in cooking. Answer options included “never”, “rarely—less than once a week”, “sometimes—once a week or more, but less than once a day”, “often—once a day or more”, “varied”, and “never”. All participants that reported at least some occasional milk consumption were further asked:


*“What type of milk was it? Was it usually …”?*


Subsequently, the NHANES inquired about several milk types, including (but not limited to) whole-milk, 1% fat milk, skim milk, and soy milk. Those participants who indicated soy milk consumption at least less than once a week were considered soy milk consumers. Those who denied soy milk consumption were considered non-consumers.

### 2.3. Covariates

Covariates for this analysis included sociodemographic data (gender, race/ethnicity, age, marital status, educational level, annual household income, household size, number of persons in the household, household food security category) as well as self-perceived general health status. Moreover, we included diabetes status (as assessed by the question: “*Have you ever been told by a doctor or health professional that you have diabetes or sugar diabetes?*”), smoking status (as assessed by the question “*Have you smoked at least 100 cigarettes in your entire life?*”), and physical activity (as assessed by the question “*In a typical week do you do any moderate-intensity sports, fitness, or recreational activities that cause a small increase in breathing or heart rate such as brisk walking, bicycling, swimming, or volleyball for at least 10 min continuously?*”). Apart from age (continuous variable) all other variables were treated as categorical variables.

### 2.4. Inclusion and Exclusion Criteria

We included all participants with the following criteria: age ≥ 20 years, available demographic data, and available milk intake data. Individuals with incomplete or missing data were not considered for this study.

### 2.5. Statistical Analysis

Statistical analysis was performed with Stata 14 statistical software (StataCorp. 2015. Stata Statistical Software: Release 14. College Station, TX, USA: StataCorp LP). The primary sampling unit variable for variance estimation and the pseudo-stratum variable as the stratification variable that were provided with both NHANES cycles were used for each analysis. To avoid missing standard errors because of strata with a single sampling unit, we used the “singleunit(scaled)” option in Stata, which is a scaled version of singleunit(certainty) and introduces a scaling factor that is derived from using the average of the variances from the strata with multiple sampling units for each stratum with a singleton primary sampling unit [45].

We used histograms and subpopulation summary statistics to check for normality of the data. Categorical variables were described with their weighted proportions and standard error in parenthesis. Normally distributed variables were described with their mean and standard error in parenthesis. All standard errors were estimated using Taylor series linearization to account for the complex NHANES sampling design. All weighting procedures were performed in accordance with the most recent applied survey data analysis techniques by Heeringa, West, and Berglund [46], and in compliance with the current National Center for Health Statistics (NCHS) data presentation standards for proportions [47]. All weighted proportions were manually screened for reliability using the user-written post-estimation Stata command “kg_nchs” [48]. Potentially unreliable proportions that did not meet the NCHS presentation standards were highlighted and clearly marked with superscript letters.

Stata’s Rao–Scott test and multivariate logistic regression models were used to examine potential associations between self-reported soy milk intake and various predictor variables. Logistic regression models were constructed based on the recommendations of Heeringa, West, and Berglund [46]. In a first step, we conducted exploratory bivariate analyses to check the eligibility of potential candidate predictors of soy milk intake. Candidate predictors of scientific interest and a bivariate relationship of significance *p* < 0.25 with the response variable were included in the multivariate logistic models. Subsequently, we evaluated the contribution of each predictor to the multivariate model using Wald tests. All variables (except age) were entered as categorical variables into the regression models. At least two models were constructed for each cycle, based on the available cycle-specific predictors. A *p*-value < 0.05 was used as the cutoff for statistical significance.

## 3. Results

The total NHANES 2015–2016 sample for analysis comprised *n* = 5264 participants with a full data set, of which *n* = 132 reported soy milk consumption. This may be extrapolated to represent *n* = 4,427,078 US Americans. The NHANES 2017–2020 pre-pandemic cycle included *n* = 8511 participants with a full dataset, of which *n* = 187 reported soy milk consumption. This may be extrapolated to represent *n* = 3,460,784 US Americans. Figure 1 shows the participant inclusion flow chart for the 2015–2016 cycle on the left side and for the NHANES 2017–2020 pre-pandemic cycle on the right side.

The weighted proportion of individuals reporting soy milk consumption in the 2015–2016 cycle was 2%, whereas it was 1.54% in the NHANES 2017–2020 pre-pandemic cycle.

### 3.1. NHANES 2015–2016

The sample characteristics of the participants reporting soy milk consumption are shown in Table 1. The weighted percentage of females consuming soy milk tended to be higher as compared to males drinking soy milk (Table 1); however, the difference was not statistically significant. Almost 43% (weighted proportion) of soy milk consumers were of Non-Hispanic White origin. Non-Hispanic Blacks and Non-Hispanic Asians accounted for more than 17% each.

Significant differences between soy milk consumers and non-consumers were found with regard to educational level. A significantly higher weighted proportion of individuals reporting soy milk intake had a college degree or higher (46.96% vs. 32.18%, *p* = 0.03). No significant intergroup differences were found with regard to household size, household food security level, and annual income. A significantly higher proportion of soy milk consumers indicated moderate recreational activities as compared to non-consumers.

In a next step, we used multivariate logistic regression models to examine potential associations between soy milk intake status (dependent variable) and various predictor variables (Table 2). While female sex did not increase the odds for soy milk consumption, Non-Hispanic Black and Non-Hispanic Asian ethnicities significantly increased the odds (OR: 2.51 and 4.87, respectively) in model 1. In a second (model 2) households with six or more persons had significantly lower odds for soy milk consumption (Table 2). Notably, said model was overall no longer statistically significant. When adding physical activity in model 3, statistical significance was retained. Participants with moderate-intensity sports and recreational activities had significantly higher odds for soy milk consumption (OR: 2.36).

### 3.2. NHANES 2017–2020

Sample characteristics of participants reporting soy milk consumption in the NHANES pre-pandemic cycle are shown in Table 3. The weighted percentage of females consuming soy milk was significantly higher in the NHANES 2017–2020 cycle: 63.45% vs. 36.55%. Only 34.55% (weighted proportion) of soy milk consumers were of Non-Hispanic White origin, whereas approximately 18.52% were Non-Hispanic Asians. Significant differences between both groups were also found with regard to educational level. The weighted proportion of individuals with a high school degree was substantially lower among soy milk consumers (16.01% vs. 27.10%, *p* = 0.006) while the weighted proportion of participants with (some) college degree tended to be higher. No significant differences were found with regard to household food security level, general (self-perceived health condition), and annual income. A significantly higher proportion of soy milk consumers indicated moderate recreational activities as compared to non-consumers. The weighted proportion of smokers also differed significantly between groups.

Again, we used multivariate logistic regression models to examine potential associations between soy milk intake status and various predictor variables (Table 4). Female sex did not increase the odds for soy milk consumption after adjustment for race/ethnicity and education level. Notably, Mexican American and Other Hispanic ethnicities significantly increased the odds (OR: 4.26 and 3.21, respectively). The same applied to Non-Hispanic Black and Non-Hispanic Asian ethnicities (OR: 2.62 and 5.60, respectively) in a second model adjusted for smoking status and moderate intensity activity. In both models, college graduates had a significantly higher OR for soy milk consumption (Table 4). The additional adjustment for physical activity did not significantly alter the findings from model 1. Participants with moderate-intensity sports and recreational activities had significantly higher odds for soy milk consumption (OR: 1.65).

## 4. Discussion

We used NHANES data to assess the prevalence of soy milk consumption in the Unites States and sought to identify potential sociodemographic predictors increasing the likelihood of its usage. The weighted proportion of individuals reporting soy milk intake in the NHANES 2015–2016 cycle was 2% and changed slightly to 1.54% in the NHANES 2017–2020 pre-pandemic cycle. Non-Hispanic Asian and Black ethnicities (as well as other Hispanic and Mexican American ethnicities in the 2017–2020 cycle) significantly increased the odds for soy milk consumption. College graduates also had significantly higher odds for consuming soy milk (OR: 2.14) in the pre-pandemic NHANES cycle. Our results also suggest that sex is apparently not an important predictor of soy milk consumption in this cross-sectional sample, while moderate physical activity was associated with higher odds.

Soy milk is one of the fastest growing categories in the U.S. plant-based non-dairy functional beverage market [49,50]. Cow milk allergies, lactose intolerance, calorie concerns, an unfavorable lipid profile, and a preference towards vegan diets for health and ethical reasons (including aspects such as environmental concerns and animal welfare) have increasingly influenced consumers across the globe towards choosing cow milk alternatives [50,51].

In addition to that, individuals are also increasingly concerned about potential negative health impacts of dairy products [52], including their high saturated fat content, their potential hormonal contamination [53], and, above all, their potential association with several diseases including various types of cancer [54,55,56]. However, recent systematic data highlighted some beneficial aspects of cow milk consumption in osteoporosis, cardiovascular diseases, and metabolic syndrome at various stages of life [57,58]. Nevertheless, concerns about acne, infant iron-deficiency anemia, prostate, colorectal and bladder cancers, and Parkinson’s disease associated with cow milk consumption remain.

For the aforementioned reasons, soy milk is as a rapidly emerging competitor to dairy milk [49]. With regard to its nutritional profile, a 2018 review suggested that soy milk is the best alternative milk for replacing cow milk in the human diet [16]. Soy milk may also favorably affect circulating estrogen levels in premenopausal women, which could reduce the risk for breast cancer [59]. In men, soy milk consumption was associated with a reduction in prostate cancer risk [60].

Despite these putative benefits, data on soy milk intake is scarce. Sociodemographic predictors and drivers of soy milk have rarely been investigated. A study by Dharmasena and Capps suggested that age, employment status, education level, race, ethnicity, region, and presence of children in a household are significant drivers of the demand for soy milk [49]. While based on a larger sample, their study dates back to the year 2008 [49]. Using more recent data from the NHANES, we were able to confirm some of the previously identified sociodemographic predictors.

Our findings may provide valuable information about soy milk consumers and could be employed in possible public health strategies to enhance soy milk product usage and consumption. Marketing for soy products is said to require meticulous consumer segmentation in order to development food products that may appeal to different populations with various opinion and tastes [61,62]. Based on our results, individuals of Non-Hispanic White ethnicity could be such a group. The same may apply to individuals with a lower education level. Targeted marketing improving the nutritional knowledge about soy milk as a potential dairy substitute could enhance consumption in said prospective buyers.

### Strengths and Limitations

The present study has various strengths and limitations that require further discussion. One major limitation is the cross-sectional nature of this analysis, which does not allow for any causal inference. Although we used a nationally-representative sample of United States Americans, the number of soy milk consumers was only modest, and some estimated reported proportions must be considered unreliable as per recent NCHS guidelines. We transparently flagged these proportions in the results section and clearly acknowledge this limitation. Furthermore, this analysis solely relied on data from the NHANES Diet Behavior & Nutrition module, it is not based on 24-h dietary recalls and does not inquire about reasons for (and barriers to) soy milk consumption. Such variables were unavailable in the employed NHANES cycles but would have significantly enriched our analysis. Finally, the NHANES “only” inquired about the usage of (soy) milk consumption as a drink or in combination with cereals. This excludes cooking and therefore some classical (vegan) meals that include soy milk, including but not limited to dairy-free macaroni and cheese, dairy-free lasagna, soy milk shakes as well as dairy-free pies, desserts, and cookies. As such, we may have underestimated the true prevalence of soy milk consumption. Nevertheless, we believe in the value of our data and call for additional studies in this particular field to enhance our understanding of soy milk consumption.

## 5. Conclusions

The weighted proportion of individuals reporting soy milk consumption in the NHANES ranged from approximately 1.54 to 2.0% in some of the latest NHNAES cycles. Several sociodemographic predictors of soy milk consumption (including race/ethnicity, household size, and educational level) were identified. Nevertheless, additional studies are warranted to gain a better understanding of drivers for (and barriers to) soy milk consumption in the United States. In light of the putative health benefits of soy milk and its more favorable environmental impact as compared to cow milk, future investigations should attempt to identify strategies that help promote its consumption.

## Figures and Tables

**Figure 1 nutrients-15-02532-f001:**
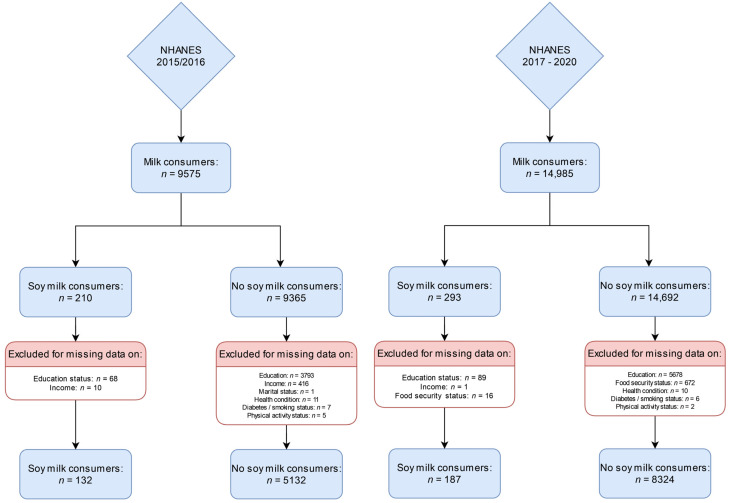
Participant inclusion flowchart for the NHANES 2015–2016 cycle (**left side**) and the NHANES 2017–2020 cycle (**right side**).

**Table 1 nutrients-15-02532-t001:** Sample characteristics by soy milk consumption status: NHANES 2015–2016.

	Soy Milk: Consumers *n =* 132	Soy Milk: Non-Consumers *n* = 5132	*p*-Value
**Sex**			*p* = 0.217 ^b^
Male	40.46% (5.80)	48.04% (0.54)	
Female	59.54% (5.80)	51.96% (0.54)	
**Age (years)**	46.40 (2.24)	47.82 (0.55)	*p* = 0.508 ^c^
**Race/ethnicity**			***p* < 0.001** ^b^
Mexican American	6.35% (2.53) ^f^	8.59% (2.05)	
Other Hispanic	8.93% (3.22) ^f^	6.10% (1.34)	
Non-Hispanic White	42.90% (7.76) ^e,f^	65.54% (3.89)	
Non-Hispanic Black	18.11% (4.27)	10.88% (2.13)	
Non-Hispanic Asian	17.06% (4.83) ^e^	5.34% (1.16)	
Other Race ^a^	6.64% (3.61) ^f^	3.56% (0.35)	
**Marital status**			*p* = 0.401 ^b^
Married/Living with Partner	60.07% (1.53) ^f^	64.69% (1.55)	
Widowed/Divorced/Separated	15.48% (0.80)	18.00% (1.19)	
Never married	24.45% (6.34)	17.31% (1.27)	
**Annual household income**			*p* = 0.315 ^b^
<20,000 US$	9.97% (2.59)	12.95% (1.23)	
>20,000 US$	90.03% (2.59)	87.05% (1.23)	
**Education Level**			*p* = 0.124 ^b^
Less than 9th grade	5.25% (1.27) ^f^	5.66% (0.91)	
9–11th grade	4.89% (2.01) ^f^	8.35% (0.90)	
High school graduate/GED ^d^	16.32% (5.07) ^f^	20.97% (1.18)	
Some college or AA degree	26.58% (4.73)	32.84% (1.52)	
College graduate or above	46.96% (5.09) ^e^	32.18% (3.09)	
**Food security category**			*p* = 0.416 ^b^
Full food security	70.91% (4.66)	71.90% (2.16)	
Marginal food security	15.32% (4.35)	10.38% (1.00)	
Low food security	8.19% (2.86) ^f^	10.73% (1.03)	
Very low food security	5.58% (2.12) ^f^	7.00% (0.58)	
**Household size**			*p* = 0.101 ^b^
One person	9.77% (2.26)	14.01% (0.81)	
Two persons	43.98% (6.72)	33.35% (1.71)	
Three persons	20.19% (3.85)	17.36% (1.38)	
Four persons	10.67% (2.62) ^e^	17.33% (1.10)	
Five persons	12.44% (4.03) ^f^	9.87% (0.69)	
Six persons	1.96% (1.15) ^e,f^	4.45% (0.57)	
Seven persons or more	0.99% (0.70) ^e^	3.63% (0.49)	
**General health condition**			*p* = 0.158 ^b^
Excellent	18.42% (0.87)	14.63% (0.87)	
Very good	32.71% (4.70)	32.62% (1.44)	
Good	35.63% (5.97)	34.66% (1.04)	
Fair	7.11% (1.50) ^e,f^	14.77% (1.09)	
Poor	6.13% (2.01) ^f^	3.32% (0.38)	
**Diabetes status**			*p* = 0.224 ^b^
Yes	7.53% (2.60) ^f^	10.84% (0.80)	
No	91.85% (2.82)	87.15% (0.84)	
Borderline	0.63% (0.43) ^e,f^	2.01% (0.30)	
**Smoking status**			*p* = 0.624 ^b^
Yes	46.37% (5.84)	43.39% (1.05)	
No	53.63% (5.84)	56.61% (1.05)	
**Moderate recreational activities**			***p* = 0.005** ^b^
Yes	65.59% (5.32) ^e^	46.72% (1.79)	
No	34.41% (5.32) ^e^	53.28% (1.79)	

Weighted proportions. Total number of unweighted observations: *n* = 5264. Continuous variables shown as mean (standard error). Categorical variables shown as weighted proportion (standard error). ^a^ = includes multi-racial; ^b^ = based on Stata’s design-adjusted Rao–Scott test, ^c^ = based on regression analyses followed by adjusted Wald tests, ^d^ = or equivalent, ^e^ = indicates significant differences in the weighted proportions, ^f^ = weighted proportions to be considered unreliable, as per recent NCHS Guidelines. Column percentages may not equal 100% due to rounding.

**Table 2 nutrients-15-02532-t002:** Multivariate logistic regression models examining potential associations between soy milk consumption status and sex, race/ethnicity, and household size.

Independent Variables	OR	CI	*p*	OR	CI	*p*	OR	CI	*p*
	Model 1			Model 2			Model 3		
**Sex**									
Female	1.34	[0.80, 2.25]	0.242	1.36	[0.81, 2.28]	0.229	1.33	[0.79, 2.24]	0.258
**Ethnicity**									
Mexican American	1.14	[0.49, 2.66]	0.750	1.44	[0.60, 3.44]	0.392	1.29	[0.54, 3.07]	0.547
Other Hispanic	2.24	[0.96, 5.23]	0.061	2.59	[1.15, 5.81]	**0.024**	2.43	[1.05, 5.63]	**0.039**
Non-Hispanic Black	2.51	[1.18, 5.40]	**0.022**	2.81	[1.38, 5.70]	**0.007**	2.72	[1.32, 5.60]	**0.010**
Non-Hispanic Asian	4.87	[2.45, 9.68]	**<0.001**	5.48	[2.74, 11.01]	**<0.001**	5.27	[2.59, 10.70]	**<0.001**
Other Race ^a^	2.87	[0.76, 10.83]	0.112	2.89	[0.75,11.21]	0.115	3.06	[0.80,11.72]	0.096
**Household size**									
1 person				0.51	[0.24, 1.11]	0.084			
3 persons				0.74	[0.41, 1.34]	0.292			
4 persons				0.40	[0.20, 0.78]	**0.011**			
5 persons				0.77	[0.30, 1.93]	0.547			
6 persons				0.26	[0.08, 0.92]	**0.039**			
7 persons or more				0.14	[0.03, 0.67]	**0.017**			
**Moderate activity**									
Yes							2.36	[1.40, 3.99]	**0.003**

Legend: ^a^ = includes multi-racial. A significant regression equation was found for model 1: F(6,10) = 4.57 (model 1) with a *p*-value of 0.017. When adding household size (model 2), the regression equation was no longer significant: F(6,10) = 4.56 with a *p*-value of 0.078. When adding physical activity to model 1 (model 3), a significant regression equation was found: F(7,9) = 8.06, *p*-value: 0.003. Reference categories were as follows: Male sex; Non-Hispanic White; Household size: two persons. Moderate recreational activities in a typical week: “no”. OR = odds ratio. CI = confidence interval. The model is based on a total *n* of 5264 participants.

**Table 3 nutrients-15-02532-t003:** Sample characteristics by soy milk consumption status: NHANES 2017–2020.

	Soy Milk: Consumers *n* = 187	Soy Milk: Non-Consumers *n* = 8324	*p*-Value
**Sex**			***p* = 0.048** ^b^
Male	36.55% (5.33) ^e^	48.09% (0.80)	
Female	63.45% (5.33) ^e^	51.91% (0. 80)	
**Age (years)**	50.26 (2.05)	48.37 (0.56)	*p* = 0.373 ^c^
**Race/ethnicity**			***p* < 0.001** ^b^
Mexican American	16.40% (4.68)	8.21% (1.12)	
Other Hispanic	12.01% (3.41)	7.40% (0.68)	
Non-Hispanic White	34.55% (5.71) ^e^	63.69% (2.44)	
Non-Hispanic Black	14.77% (3.22)	11.24% (1.43)	
Non-Hispanic Asian	18.52% (2.92) ^e^	5.52% (0.84)	
Other Race ^a^	3.75% (1.60) ^f^	3.95% (1.60)	
**Marital status**			*p* = 0.430 ^b^
Married/Living with Partner	56.74% (3.62)	61.82% (1.34)	
Widowed/Divorced/Separated	23.05% (4.07)	18.92% (0.76)	
Never married	20.21% (3.92)	19.26% (1.09)	
**Education Level**			*p* = 0.080 ^b^
Less than 9th grade	5.81% (1.29) ^f^	3.64% (0.36)	
9–11th grade	6.95% (1.82)	7.12% (0.33)	
High school graduate/GED ^d^	16.01% (3.61) ^e^	27.10% (1.38)	
Some college or AA degree	32.09% (5.44)	30.56% (0.92)	
College graduate or above	39.13% (4.84)	31.57% (2.14)	
**Food security category**			*p* = 0.304 ^b^
Full food security	65.45% (4.41)	72.22% (1.14)	
Marginal food security	14.61% (2.98)	10.73% (0.58)	
Low food security	11.70% (2.31)	10.45% (0.61)	
Very low food security	8.24% (2.13)	6.59% (0.48)	
**General health condition**			*p* = 0.285 ^b^
Excellent	17.43% (3.79)	13.94% (1.05)	
Very good	24.71% (4.84)	32.34% (0.87)	
Good	40.38% (3.95)	35.15% (0.96)	
Fair	13.77% (3.40)	16.01% (0.74)	
Poor	3.70% (1.14) ^f^	2.57% (0.15)	
**Ratio of family income to poverty**			*p* = 0.443 ^b^
<1	11.68% (2.24)	11.82% (0.84)	
≥1 and <2	19.61% (3.91)	17.86% (0.87)	
≥2 and <3	18.48% (3.44)	14.19% (0.80)	
≥3	50.23% (5.60)	56.12% (1.57)	
**Diabetes status**			*p* = 0.289 ^b^
Yes	15.11% (2.96)	11.61% (0.42)	
No	81.86% (2.95)	85.90% (0.41)	
Borderline	3.03% (0.92) ^f^	2.49% (0.29)	
**Smoking status**			***p* = 0.010** ^b^
Yes	29.22% (4.25) ^e^	42.59% (1.22)	
No	70.78% (4.25) ^e^	57.41% (1.22)	
**Moderate recreational activities**			***p* = 0.009** ^b^
Yes	58.62% (4.21) ^e^	46.64% (1.17)	
No	41.38% (4.21) ^e^	53.36% (1.17)	

Weighted proportions. Total number of unweighted observations: *n* = 8511. Continuous variables shown as mean (standard error). Categorical variables shown as weighted proportion (standard error). ^a^ = includes multi-racial; ^b^ = based on Stata’s design-adjusted Rao–Scott test, ^c^ = based on regression analyses followed by adjusted Wald tests, ^d^ = or equivalent, ^e^ = indicates significant differences in the weighted proportions, ^f^ = weighted proportions to be considered unreliable, as peer recent NCHS Guidelines. Column percentages may not equal 100% due to rounding.

**Table 4 nutrients-15-02532-t004:** Multivariate logistic regression models examining potential associations between soy milk consumption status and sex, race/ethnicity, education level, and age.

Independent Variables	OR	CI	*p*	OR	CI	*p*
	Model 1			Model 2		
**Sex**						
Female	1.57	[0.97, 2.56]	0.067	1.55	[0.96, 2.51]	0.071
**Ethnicity**						
Mexican American	4.26	[2.07, 8.76]	**<0.001**	4.16	[2.52, 10.18]	**<0.001**
Other Hispanic	3.21	[1.61, 6.41]	**0.002**	3.22	[1.85, 6.91]	**0.002**
Non-Hispanic Black	2.55	[1.56, 4.17]	**0.001**	2.62	[1.69, 4.44]	**<0.001**
Non-Hispanic Asian	5.70	[3.82, 8.53]	**<0.001**	5.60	[4.23, 9.30]	**<0.001**
Other Race ^a^	1.85	[0.73, 4.72]	0.185	1.95	[0.78, 4.99]	0.150
**Education level**						
Less than 9th grade	1.51	[0.86, 2.65]	0.143	1.60	[0.93, 2.77]	0.087
9–11th grade	1.36	[0.78, 2.35]	0.265	1.47	[0.85, 2.57]	0.159
Some college or AA degree	1.83	[0.93, 3.62]	0.079	1.75	[0.88, 3.47]	0.105
College graduate or above	2.14	[1.21, 3.80]	**0.011**	1.84	[1.01, 3.33]	**0.045**
**Moderate activity**						
Yes				1.65	[1.14, 2.40]	**0.011**
**Smoking**						
Yes				0.82	[0.51, 1.32]	0.402

Legend: ^a^ = includes multi-racial. Significant regression equations were found for both models: F(10,16) = 21.98 (model 1) and F(12,14) = 21.16 (model 2), respectively, with a *p*-value < 0.001 for both. Reference categories were as follows: Male sex; Non-Hispanic White; High school graduate/GED; Moderate recreational activities in a typical week: “no”; Smoking: “no”. OR = odds ratio. CI = confidence interval. The model is based on a total *n* of 8511 participants.

## Data Availability

Data are publicly available online (https://wwwn.cdc.gov/nchs/nhanes/Default.aspx; accessed on 2 July 2022). The datasets used and analyzed during the current study are available from the corresponding author upon reasonable request.

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
