# Peer review of "Soy Milk Consumption in the United States of America: An NHANES Data Report"

_nutrients, 2023, doi:10.3390/nu15112532_

Round 1

Reviewer 1 Report

The authors provide a robust analysis of the soy milk consumption data in the NHANES dataset. The results are only of moderate interest and unsurprising, but there are no major problems with the study that should prevent its publication.

I have some suggestions to improve the value of this paper to the reader:

- Clarification around the growth of soy/plant-based diets consumption. The authors assert in many places through the manuscript that e.g. "plant-based diet is increasingly adopted", "increase in the demand for plant-based foods", etc. However, it is never stated in what environment this is meant: do the authors mean globally? In the US? My understanding is that this phenomenon is mostly confined to high-income countries, particularly in Europe and North America, with the opposite true in many developing economies. Suggest clarifying and evidencing these broad statements. L34, 47, 280

- Even-handedness on environmental and health impacts of soy and dairy. Currently, the authors almost solely discuss the beneficial aspects of soy for health and environmental impact, and almost solely the negative health and environmental impacts of dairy. What is included does not need correcting, but the current arrangement leaves the reader wondering whether the paper was funded by the soy industry! (I see that it wasn't). I suggest including some material on the environmental harms of soy (e.g. 10.1021/acs.est.9b06874 ; 10.1016/j.jclepro.2021.130182), the areas where it is not as rich a nutrient source as dairy (e.g. calcium, zinc; L58-66), and discussing the beneficial health aspects of dairy consumption (e.g. 10.1186/s12986-020-00527-y ; 10.1080/07315724.2018.1491016; 10.1093/advances/nmz020; L284). This would provide a more balanced narrative.

- The reference numbering appears to be wrong. e.g. L48, I think you mean ref 13 here, not 14. Requires fixing and is potentially present throughout.

- 2% to 1.54% is a very small change, but the authors on several occasions talk about this as a decrease. I think your phrasing in the Conslusions: "ranged from approximately 1.54 to 2.0 %", is more accurate. Given the uncertainties around the NHANES data and the small proportion of the sample that consumed soy, it is potentially misleading to assert that there was a decrease. It would be safer to say "consumption changed little between the two cycles".

Minor comments:

- cow milk or cow's milk. You use both in the article, suggest selecting one for consistency.

- "plant-based beverage" is a term also used in the literature, suggest adding as a keyword

- clarify what you mean by plant-based diets, plant-based foods, and plant-based alternatives. You use all three in this manuscript, but given the variation in their meaning to different people, clarification is merited. Something from your own work in ref 2, perhaps.

- L78: however, plant-based beverages usually score lower than cow's milk in sensory work (e.g. 10.1016/j.foodqual.2022.104599)

- L155: reference brackets missing

- Figure 1 and 2: would it not be more visually appealing for these to be combined into a single figure? Easier for the reader to compare the two then as well - with the current arrangement they may end up 3 pages apart in the final publication.

- L215: = sign instead of )

- L279: "animal welfare" would be a broader and more scientific term than "animal cruelty"

Very good. A couple of suggestions:

L59: I don't think "elective" is the right word here. "Nutritional" would be more explicit.

L317: I think the phrasing is wrong here. Do you simply mean "...would have significantly enriched our analysis."

Author Response

Dear Reviewer,

We would like to thank you and the reviewers very much for careful and thorough reading of this manuscript and for the thoughtful comments and constructive suggestions, which help to improve the quality of this article. We made all the requested revisions to our original manuscript based on all the comments we received from you. All changes have been clearly marked in yellow and blue color. Please find our specific point-by-point response attached.

Sincerely yours,

The authors

Reviewer 2 Report

The topic of the manuscript (nutrients-2428317-peer-review-v1) is an interesting because the adoption of plant-based diets affects people’s quality of life. In the study, the authors used data from the National Health and Nutrition Examination Surveys to assess soy milk usage in the Untited States, and identified potential predictors for its consumption in the US general population. Regarding results, some concepts should be carefully revised. There are many problems that have not been explained clearly.

The questions are too simple and don’t cover lifestyle. The results do not reflect the real reasons for consumption. Others, the quality of figures should be improved.

Author Response

(The authors gave the same response as above.)

Round 2

Reviewer 2 Report

The author made careful revisions.